# Systematic Development, Validation and Optimization of a Human Embryo Culture System

**Mitchel C. Schiewe [1,\*], Shane Zozula [1], Nancy L. Nugent [1], John B. Whitney [1], Ilene Hatch [2], C. Terence Lee [3] and Robert E. Anderson [4]**

[1] Ovation Fertility, Newport Beach, CA 92663, USA; szozula@ovationfertility.com (S.Z.); nnugent@ovationfertility.com (N.L.N.); jwhitney@ovationfertility.com (J.B.W.)
[2] Fertility Center of Southern California, Irvine, CA 92604, USA; ihatchmd@gmail.com
[3] Fertility Center for Assisted Reproduction & Endocrinology, Brea, CA 92821, USA; Tlee@FCARE.com
[4] Southern California Center for Reproductive Medicine, Newport Beach, CA 92663, USA; rea@socalfertility.com
[\*] Correspondence: mschiewe@ovationfertility.com; Tel.: +949-642-5954

**Abstract:** Objective: To develop and validate a reliable in vitro culture system for human embryos. Design: Retrospective analyses of a series of four studies were conducted between 2006 and 2010 to assess the effect of incubator type ($CO_2$ box versus Tri-gas minibox), media type, oil type, and hyaluronate supplementation. Optimization of in vitro blastocyst development was verified by assessing our National CDC/ART Surveillance reports between 2010 and 2016. Material and Methods: All patients experienced controlled ovarian hyperstimulation, followed by egg retrieval 35 h post-hCG. Cumulus-oocyte complexes were temporarily cultured in P1 or LG Fert medium plus HSA. Eggs were moved to a more complex media (G-medium or Global®-LG medium) containing a synthetic protein and embryo adhesion supplement (SPS and EAS, respectively; mLG) post-ICSI insemination. Zygotes were assigned to group culture in 25 µl droplets under oil (light mineral oil or paraffin oil; 37 °C) and embryo development was evaluated on Days 3, 5, and 6 and transferred on Day 3 to 5 depending on the number/quality of embryos available and the IVF history of the patient. Transfers were performed under ultrasound guidance, primarily using a Sureview-Wallace catheter, and enriched ET medium containing 500 µg/mL EAS. Results: Pilot study results (Expt. 1) showed that a mLG single-step medium could be effectively used in combination with Sanyo MCO-5 tri-gas (TG) incubators. Once adapted to SCIRS Lab in 2007 (Expt. 2), the latter culture system yielded improved blastocyst production and pregnancy outcomes compared to $CO_2$ in air sequential incubation in P1/Multi-blast medium. In Expt. 3, the mLG/TG system yielded high levels of ≥2BB quality blastocysts (51 to 66%) across all age groups, and greater ($p < 0.05$) pregnancy success/live birth rates using fewer embryos transferred on Day 5 versus Day 3. After validating its clinical effectiveness, mLG was then prospectively compared to a new generation G-media (1.5 & 2.5; Expt. 4) and determined that the crossover treatment using paraffin oil (Ovoil™) allowed the mLG system to be optimized. Subsequently, a compilation of our Annual CDC/ART reported data over six years verified the overall viability of in vitro cultured and vitrified blastocysts produced in the mLG/TG system. Conclusion: By systematically evaluating and implementing various components of an embryo culture system we were able to optimize blastocyst development over the last decade. Our mLG/TG culture system modified an exceptionally well designed KSOM[AA] LG medium using endotoxin-free EAS and SPS additives to support cellular membrane wellness under stressful in vitro conditions (e.g., culture, cell biopsy, vitrification). Our use of the mLG/TG culture system has proven to be effective, creating reliably high blastocyst production, implantation, and healthy live births.

**Keywords:** embryo culture; media; blastocyst; implantation; Global®; hyaluronate

---

## 1. Introduction

The evolution of todays' embryo culture systems spans six decades, beginning with the pioneering contributions of Wesley Whitten [1], Ralph Brinster [2], John Biggers [3], and David Whittingham [4], to name a few who sought to understand the physiology and metabolism of mammalian embryos, as reviewed in depth previously [5–7]. Other early pioneers developed complete media for mammalian cell cultures emphasizing the beneficial role of amino acids [8] and protein fractions [9]. Yet, it was Whitten and Biggers [10] who first assessed complete mouse embryo development in a simple chemically defined medium. Two decades later, Henry Leese and his associates generated a wealth of knowledge regarding reproductive tract physiology and the physiochemical pathways of embryo metabolism [11,12], prompting more insight into media formulation. Meanwhile, Patrick Quinn and coworkers [13] had developed a simple medium based on human oviductal fluid to reliably grow cleavage stage embryos, whereas early investigators who strove to reliably grow human blastocysts used cell co-cultures to gain the necessary beneficial factors to promote post-compaction human embryo development [14,15]. It was Biggers and Leese who mentored the next generation of reproductive biologists that created the defined media and macromolecular products used today.

While Lawitts and Biggers [16] were developing a simplex optimization of medium (SOM), David Gardner was focusing on the relationships of embryo and maternal physiology [17] to meet the physiochemical needs of the embryo pre- and post-compaction. In doing so, he and his colleague, Michele Lane, developed the concept of sequential media [18–20]. Of paramount importance to optimizing embryo culture in vitro was a need to minimize embryo stressors. They evaluated the importance of amino acids as osmolytes and gene regulators [21,22], beneficial paracrine effects of group culturing [23,24], and the negative impact of ammonium resulting from L-glutamine metabolism [25,26]. Another critical source of embryo stress is the presence of reactive oxygen species. It has been shown that excessive peroxidation can occur when light mineral oil is exposed to UV light, but not to extended exposure to 37 °C [27,28]. On the other hand, paraffin oil has few unsaturated hydrocarbons and is more resistant to oxidative reactions [29]. Overall, detailed review of the justification, development, and validation of sequential media is summarized elsewhere [18,30–32].

As reviewed by Summers [33], Biggers took a different approach in which he provided a balance of ingredients for embryo utilization at different points of development. The addition of potassium and 19 amino acids (essential and non-essential as described by Eagle) to SOM, forming KSOM$^{AA}$, improved expanded blastocyst formation and markedly increased cell numbers, especially inner cell mass. The latter KSOM$^{AA}$ became the formulation of Global® medium, and, with glycyl-L-glutamine replacing L-glutamine [34,35], LifeGlobal produced a more stable complete medium with a longer shelf life and reduced ammonium production overtime, similar to alanyl-L-glutamine supporting G-series sequential media [26]. As a single-step embryo culture medium, Global® (LG) offered greater ease of use compared to sequential media systems in respect to incubator management, pH/Lot variations, and culture/ET dish set-up/patient. As a complete medium, however, LG lacked one intriguing, innovative additive found in Vitrolife G-Series media, hyaluronan. Hyaluronan (HA), also referred to as hyaluronic acid or hyaluronate, is a high molecular weight macromolecule found in all vertebrate tissues and body fluids. It is synthesized in the plasma membrane of cells and has important physiochemical and physiological functions [36]. More specifically, at concentrations of 0.125–0.5 mg/mL HA under in vitro embryo culture conditions it was found to promote trophectodermal cell proliferation, adherence, migration, and implantation [37] in the mouse model; increase blastocyst development in pigs [38]; and improve pregnancy outcomes in humans [39].

The overall goal of this study was to validate that a modified LG medium (mLG) containing an embryo adhesive supplement (EAS; recombinant HA) could reliably support viable in vitro blastocyst production. Secondly, we evaluated different culture conditions (gas environment, oil type) in an attempt to optimize a culture system that supports excellent blastocyst viability with proven live births. Finally, this study strived to verify the consistency of high levels of pregnancy success in all age groups following years of fresh embryo and cryopreserved blastocyst transfers.

## 2. Materials and Methods

*2.1. Experimental Design and Objectives*

### 2.1.1. Informed Patient Consent

All patients involved in the validation studies in Phase 1 signed informed General IVF consents with the knowledge that "the ART Lab may conduct clinical comparisons in standard procedures aimed to make potential improvements in patient outcomes without any additional risk to the patient, their gametes, or embryos". Overall, the research conduct, ethics, and analyses (retrospective) of this study were approved for publication by an institutional research ethics committee (OREC, #OF191220A, 12-20-2019).

### 2.1.2. Phase 1. Development and Validation of Embryo Culture Systems

**Expt. 1.** Ongoing clinical pregnancy rates in an unselected IVF population (FSAC Lab, Thousand Oaks, CA) were retrospectively compared using two different media formulations (Vitrolife $G_{1.3}/G_{2.3}$ sequential media and LifeGlobal{LG} Global® single-step medium) under incubation conditions of 6% $CO_2$ in air, which were subsequently contrasted with LG medium under a 6% $CO_2$/5% $O_2$/89% $N_2$ gas incubator environment using two different incubator types (miniature Sanyo MCO-5M vs Big box-Hearus 240) (Figure 1). Over a 1-year period between April 2006 and March 2007, treatments were applied in quarterly increments ($n$ = 148 patients; including 19 donor egg cycles, 12.8%). The purpose was to justify and help decide on an embryo culture system (media and incubator) that could be adapted at the SCIRS Lab (Newport Beach, CA, USA).

**Expt. 2.** Following a six-week set-up period consisting of heated out-gassing, aseptic cleaning/sterilization, and quality control checks, 12 Sanyo tri-gas incubators (3 per stack) were implemented into routine clinical use in September 2007. Ongoing clinical pregnancy rates in an unselected IVF population ($n$ = 389 patients; including 46 donor egg cycles, 11.8%) were retrospectively contrast to the previous culture system (BigBox-$CO_2$ in air; Irvine Sci. P1/Multi-Blast sequential media) over a similar 6-month interval. Our aim was to validate the clinical use of the miniaturized, tri-gas (TG) Sanyo MCO-5M (Panasonic Healthcare NA, Bensenville, IL, USA) incubators in human IVF using our modified LG (mLG)/TG culture system.

**Expt. 3**. Continuing to evaluate the performance and merits of the mLG/TG culture system, a retrospective analysis of 524 IVF cycles ($n$ = 69 involving donor eggs, 13%) was conducted to evaluate differences in clinical pregnancy and live birth outcomes between Day 3 cleavage embryo transfer (ET) and Day 5 blastocyst ET. Our purpose was to validate the overall effectiveness of our blastocyst culture system.

**Expt. 4**. A prospective comparison of G1/2.5 sequential and mLG media systems was conducted. In 2009, 90 autologous patient cycles were enrolled in a randomized comparative trial to determine if we could improve clinical outcome with the next generation G5 Series™ media. Patients were sorted by age and randomly assigned to LG or G5 Series™ the day before OPU. Experimental group A (≤34 years old) and B (35–40 years old) contained patient oocyte sources possessing >10 follicles. Cumulus oocyte complexes (COCs) were temporarily held in P1 (LG) or GIVF (G5 Series™) medium + 5% HSA until ICSI was performed. Post-ICSI, all oocytes were cultured in either mLG medium + 7.5% SSS and recombinant hyaluronate (125 μg/mL; similar to G-media) or G5 Series™ + 10% SSS in 25 μL droplets in 60 × 15 mm Nunc culture dishes under light mineral oil (7.0 mL Global®) or paraffin oil (Ovoil™), respectively. In a secondary crossover experiment (4B), mLG microdroplets were overlaid with Ovoil™. Tri-gas mini Sanyo (MCO-5) incubators were used, varying the %$CO_2$ each lot to achieve the desired media pH for LG 7.31–7.35 and G5 Series™ 7.25–7.29. Normal 2PN zygotes and viable unfertilized eggs were isolated into a new culture dish on Day 1 (mLG/G1) and on Day 3 (mLG/G2) following embryo morphology evaluations.

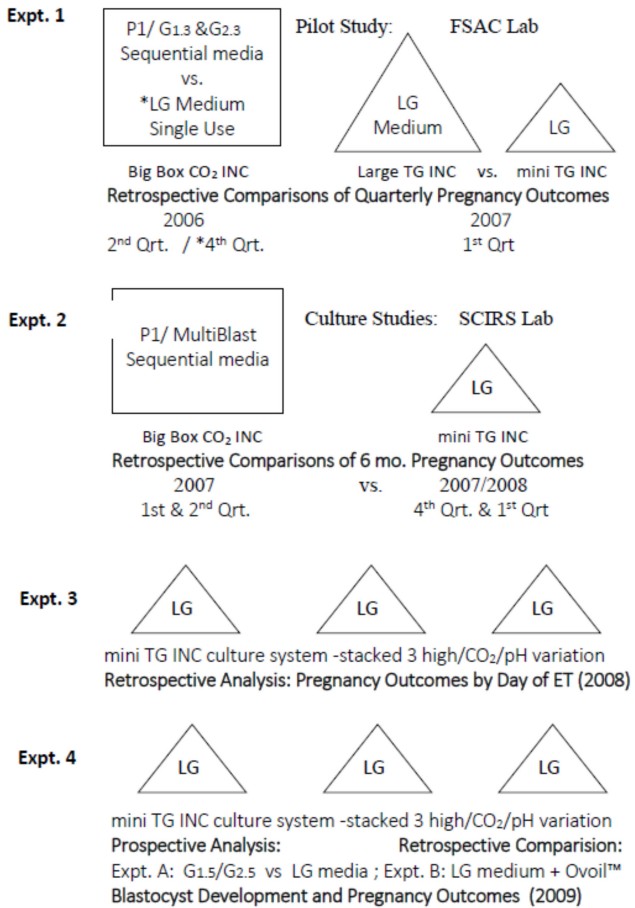

**Figure 1.** Flow chart of the Phase 1 experimental design of the systematic development, implementation and validation of a human embryo culture system. The study strived to integrate and prove the optimized effectiveness of a single-step medium (Global®; LifeGlobal, LG) culturing embryos in miniaturized tri-gas (TG) incubators (INC; Expt. 1, 2 and 3), as opposed to larger (Big box) $CO_2$ in air INC (Expt. 1 and 2). Additionally, the studies contrasted pre-existing sequential media systems (Expt. 1 and 2) and concluded with a final prospective comparison of LG to an improved Vitrolife G-media (G1.5 and G2.5, Expt. 4A) with a secondary crossover experiment (4B) proving that Ovoil™ was the key Vitrolife component needed to optimize our modified LG medium (+ hyaluronate; mLG) under TG conditions.

### 2.1.3. Phase 2. Verification of an Optimized Embryo Culture System

**Expt. 5**. Cumulative implantation, pregnancy, miscarriage and live birth success across age groups was analyzed from the annual CDC/ART Surveillance reports of three physicians using the SCIRS/OF-NB Lab over the last 15 years. This retrospective analysis aimed to verify the overall viability of blastocysts produced in the mLG/Ovoil™/TG embryo culture system. We began integrating blastocyst biopsying/PGT-A analysis into our laboratory by 2013 and transitioned into freeze all cycles by 2014. Therefore, to best characterize the quality/viability of blastocyst production in the Phase 2 verification data, it is expressed as Fresh ET cycles from 2010–2013 ($N = 1218$) and frozen embryo transfer (FET) cycles between 2013–2016 ($N = 1808$). Approximately 12% of these cycles included donor egg use.

### 2.2. Statistical Analysis

Chi squared analyses were performed to contrast differences ($p < 0.05$) in pregnancy outcomes (e.g., clinical pregnancy, implantation, and live birth rates). Although student's *t*-test was used to assess the significance of potential patient population differences in previously published group comparisons

(e.g., average age, stimulation protocol, average follicle count, and day 3 FSH; [40,41]), no attempt has been made in this paper to rule out possible difference among patient groups due to the historic, retrospective nature of the comparisons. Population differences were considered negligible as we were using unselected patient groups (as annually reported to the Centers for Disease Control, CDC) to assess trends and relationships. Our goal was not necessarily to imply one treatment was better than the next, but instead to validate the effectiveness of an application being similar to and greater than a standard method. Our overall aim was to validate the reason for change and justify it based on possible improvements that would lead to an optimization of our embryo culture system.

### 2.3. General ART Applications

All patients experienced controlled ovarian hyperstimulation, followed by egg retrieval 35 h post-hCG. Cumulus-oocyte complexes (COCs) were recovered and placed in simple basic salt starter medium (P1 medium, Irvine Sci., Santa Ana, CA, USA; or LG Fert medium, LifeGlobal, Guilford, CT, USA) + 5% HSA until insemination. The majority (99%) of IVF cycles involved the use of intracytoplasmic sperm injection, as previously detailed by Schiewe [42]. The COCs were briefly exposed to a warm (37 °C) hyaluronidase solution (80 u/mL, Irvine Sci., Santa Ana CA, USA) for 1–2 min and repeatedly pipette through a Pasteur pipette. Corona radiata surrounded oocytes were then isolated in LG-Hepes buffered medium (LG-H) + 10% SPS and further stripped free of their granulosa cells with serial flexipettes (300-170-140 μm ID). Mature oocytes underwent ICSI 2–6 h post-retrieval. Following the fertilization check, zygotes and unfertilized eggs (0PN) were moved to a more complex media containing either 5% HSA (G-media) or 7.5% SPS (SS, Irv. Sci. or LGPS, LifeGlobal) + an embryo adhesive supplement (EAS: 125 μg/mL recombinant sodium hyaluronate; stock solution = 5 mg/mL, tested endotoxin-free <0.05 units) to our Global$^®$ medium (mLG). Note that annual selection and usage of high quality HSA and SPS lots was based on selecting the lowest available endotoxin levels (preferably undetected LAL level), with at least an annual supply obtained. Embryos were grouped cultured in 25 μl droplets (up to 8 embryos /droplet) in 60 × 15 mm Nunc culture dishes under oil (7 mL, Global$^®$ light mineral oil or Ovoil$^®$). All oocyte and embryo handling procedures performed under ambient air conditions involved the transient use of LG-H medium.

### 2.4. Incubator Conditions and Quality Control

The gas environment of all tri-gas (TG) incubators was supplied by a single 100% $CO_2$ tank (H-type), and a 176L $LN_2$ vapor tank, both using gas guard back-up systems. $CO_2$ tanks were typically replaced prior to reaching 500 psi, as to avoid aspiration of possible chemically active contaminants. Incubators were set to 5.0 to 6.5% $CO_2$/5% $O_2$/balance $N_2$ gas, depending on achieving a desired pH with different media Lot #'s. We performed monthly titration curves at 5.0, 5.5, 6.0, and 6.5% $CO_2$ to assess pH levels. $CO_2$ set points were adjusted to achieve a pH of 7.3–7.32 for fertilization to Day 3 embryo growth and 7.35–7.37 for Day 3 through blastocyst development (Day 6). Daily laboratory quality control practices were implemented to record incubator temperatures and assess $CO_2$ level using a calibrated gas analyzer. If $CO_2$ drifted more than 0.02% in more than 2 days, the incubator $CO_2$ was recalibrated.

### 2.5. Embryo Assessments and Procedural Manipulations

All embryos underwent standard quality grading of Day 3 to assess cell number, symmetry, and fragmentation and assigned an overall grade of A, B, or C (good to poor, respectively). All blastocysts were graded on Day 5 and 6 according to Gardner criteria [20] accounting for stage (1–6, early BL to hatched BL, respectively) and quality of the inner cell mass (ICM) and trophectoderm (TE) being assigned A, B, or C dual grades. Blastocyst formation was documented and typically only those possessing ≥2BB grade were cryopreserved or may have first underwent TE biopsy procedures for PGT-A determination if ≥3BB grade, as previously detailed [41,43]. All fair to excellent quality blastocysts were vitrified using our microSecure vitrification (μS-VTF) procedure [44,45]. Embryos were

transferred on Day 3 or 5 depending on the number/quality of embryos available and the IVF history of the patient. Embryos were exposed to EAS supplemented mLG media (500 µg/mL) prior to ET at least 1 h prior to transvaginal ultrasound guided ET [46], primarily using a Sureview-Wallace soft catheter. Patient ET and post-ET hormonal support, as well as pregnancy monitoring, has been the subject of prior published investigations in our lab [40,46,47].

## 3. Results

Pilot studies contrasting media and incubation culture systems in Expt. 1 revealed that sequential G-media under $CO_2$ in air created higher pregnancy outcomes than LG single use medium in younger women, except when the 35–37 year old population was analyzed (Table 1). Meanwhile, the use of TG incubation with the mLG medium proved beneficial, especially in the mini-box Sanyo incubators. The combined TG pregnancy data tended ($p < 0.10$) to be higher than the prior $CO_2$ in air data for both age groupings. Cumulative data analysis of $CO_2$ in air exhibited a reduced ($p < 0.05$) live birth rate (54%) mitigated by a higher ($p < 0.05$) SAB rate (22%) compared to the TG systems (63% and 8.2%, respectively; Table 1) applied at FSAC.

**Table 1.** Expt. 1: Preliminary investigation retrospectively analyzing the clinical effectiveness of two distinct embryo culture systems (sequential G3 media versus single-step mLG medium), as well as the early validation of MCO-5 tri-gas (TG) Sanyo incubators.

| Categories | $n$ [δ] | # (%)+β-hCG | # (%) Live Birth | $n$ | # (%)+ß-hCG | # (%) Live Birth |
|---|---|---|---|---|---|---|
| | | AGE <35 yo + Donor Cycles | | | AGE <38 yo + Donors | |
| | | 6% $CO_2$/Air Incubation | | | | |
| | | [2ndQrt06] | | | | |
| P1/G1.3/2.3 media | 49 (7) [δ] | 40 (82) * | 31 (63) * | 75 | 54 (72) | 42 (56) |
| | | [4thQrt06] | | | | |
| mLG medium ** | 49 (5) [δ] | 29 (59) | 24 (49) | 65 | 42 (65) | 33 (51) |
| | | 6% $CO_2$/5% $O_2$/89% $N_2$ Incubation {mLG media} | | | | |
| | | [1stQrt07] | | | | |
| Sanyo Mini | 29 (4) [δ] | 22 (76) | 21 (72) * | 38 | 25 (66) | 24 (63) |
| Heraeus 240 | 21 (3) [δ] | 15 (71) | 13 (62) | 32 | 24 (75) * | 20 (63) |
| Combined TG | 50 (7) [δ] | 37 (74) | 34 (68) | 70 | 49 (70) | 44 (63) |

yo = years old; +β-hCG = positive pregnancy; δ—number in parenthesis represents donor egg cycles. * Indicates a difference ($p < 0.05$) within column subsection; ** mLG = Global medium supplemented with 125 µg/mL recombinant hyaluronate + SPS.

Transferring the technology and knowledge gained in the FSAC Lab, a dedicated miniature Sanyo TG incubator system was established at the SCIRS Lab in June 2007. Upon evaluating the initial clinical effectiveness over a 6-month interval, high normal ICSI fertilization rates (75–80% 2PN) were attained. Furthermore, while high quality embryo production was similar to the previous $CO_2$ in air/sequential culture system, a trend to higher ($p < 0.10$; Table 2) live births occurred in women under 41 years old. The latter observation was considered significant, as the combined biochemical pregnancy/spontaneous miscarriage rate was lower ($p < 0.05$) in the mLG/TG system (8.7%) compared to the 21.8% using the traditional $CO_2$ in air incubator culture system.

Continued analysis of the potential advantages of the single use mLG/TG system for routine blastocyst production was clearly evident in Expt 3 (Table 3). In 2008, 6956 human oocytes were surgically recovered from COH patients. High levels of good-excellent quality BLs were produced (51 to 66%) independent of age. This ability to routinely produce blastocyst stage embryos offered physicians better embryo selection at the time of transfer. Higher pregnancy rates ($p < 0.05$) were then observed across all age groups transferring fewer embryos when performing Day 5 vs Day 3 ETs. Clinical pregnancy loss by Day of ET, within age groups, was similar ($p > 0.1$), except in women aged 38 to 40 years old, who experience twice as many early pregnancy losses (42.4% vs 19%) based

on Day 3 embryo selection. Patients performing blastocyst ET ($n$ = 242; 46%) produced a mean of 17.8 eggs/patient compared to 9.4 eggs for D3-ET patients, which does represent a population bias.

**Table 2.** Expt. 2: Retrospective comparison of a sequential big-box incubator $CO_2$ only culture system to a single use mLG medium miniature-box TG culture environment.

| Group | INC | *N* | Day 3 8–10 cell/A | Day 5/6 BL/AA-BB * | Mean # ET | # (%) + β-hCG | # (%) Live Birth |
|---|---|---|---|---|---|---|---|
| Donor Egg | TG | 29 | 62.80% | 60.30% | 2.28 | 25 (86.2%) | 22 (75.9%) |
| | $CO_2$ | 17 | 72.30% | 14.50% | 2.65 | 13 (77%) | 11 (65.0%) |
| ≤ 34 yo | TG | 69 | 64.90% | 53.90% | 2.63 | 53 (76.8%) | 50 (72.5%) |
| | $CO_2$ | 65 | 65.70% | 31.50% | 3.03 | 47 (72.3%) | 38 (58.5%) |
| 35–37 yo | TG | 38 | 59.70% | 47.00% | 2.95 | 25 (65.8%) | 23 (60.5%) |
| | $CO_2$ | 47 | 69.30% | 15% | 3.77 | 23 (49%) | 19 (40.4%) |
| 38–40 yo | TG | 49 | 65.70% | 51.40% | 3.39 | 24 (46.9%) | 21 (42.9%) |
| | $CO_2$ | 17 | 82.90% | 15.80% | 3.96 | 18 (35.3%) | 11 (21.6%) |
| 41–43 yo | TG | 41 | 60.00% | 25.50% | 2.8 | 13 (31.7%) | 7 (17.1%) |
| | $CO_2$ | 17 | 60.00% | 13.70% | 4.03 | 9 (29%) | 6 (19.4%) |

* Blastocysts graded as freezable or transferable: % estimate is not corrected for Day 3 ET's; $CO_2$ = 6× $CO_2$ gas Forma Incubators; P-1 / MultiBlast sequential culture medium; TG =12× mini Sanyo MCO-5 tri-gas incubators; mLG = single use Global medium; supplemented with 125 µg/mL recombinant sodium hyaluronate + 7.5% LGPS.

**Table 3.** Expt. 3: Validation of the mLG medium/Sanyo MCO-5 tri-gas incubation system aimed to optimize Day 5 blastocyst culture and embryo transfer.

| Pt. Age, ET | #Pts | µ #/ET | +b-hCG (%) | Clinical Preg. (%) | Live Birth (%) | # BL AA-BB (%) |
|---|---|---|---|---|---|---|
| Donor, D5 | 53 | 2.2 | 41 (77) | 38 (72) | 36 (68) * | 469/711 (66) |
| ≤34yo, D5 | 88 | 2.1 | 64 (73) * | 57 (65) * | 48 (55) * | 602/1039 (58) |
| 35–37yo, D5 | 49 | 2.5 | 34 (69) * | 32 (65) * | 29 (59) * | 312/529 (59) |
| 38–40yo, D5 | 40 | 3.1 | 28 (70) * | 21 (53) * | 17 (43) * | 216/423 (51) |
| 41–43yo, D5 | 12 | 4.0 | 9 (75) * | 7 (58) * | 3 (25) * | 57/101 (56) |
| Donor, D3 | 16 | 2.7 | 13 (81) | 12 (75) | 9 (56) | ND |
| ≤ 34yo, D3 | 61 | 3.2 | 35 (57) | 33 (54) | 26 (43) | ND |
| 35–37yo, D3 | 63 | 3.4 | 32 (51) | 29 (46) | 22 (35) | ND |
| 38–40yo, D3 | 76 | 3.6 | 43 (57) | 33 (43) | 19 (25) | ND |
| 41–43yo, D3 | 66 | 3.3 | 16 (24) | 14 (21) | 6 (9) | ND |

* Different ($p$ < 0.05) than the corresponding Day 3 transfer group. ND = not determined.

We concluded our Phase 1 experimental analysis with a prospective, randomized comparative study which revealed the Vitrolife G1.5/2.5 series™ medium with Ovoil™ significantly enhanced live birth outcomes in the younger patient age group (<35 yo; Table 4) in contrast to mLG, due to an appreciable increase in early losses. Meanwhile good quality blastocyst production was actually higher ($p$ < 0.05) in the mLG group. No differences were observed in the 35–40 years old subsample of good prognosis patients (Table 4). Interestingly, when Vitrolife Ovoil™ was used to overlay LG microdroplets in Expt. B, even higher ($p$ < 0.05) blastocyst development occurred (Table 4). More importantly, an increased ($p$ < 0.05) live birth rate was observed, being similar to the initial G5 data in Expt. A.

By the end of 2009, it had been determined in Expt. 4 that the mLG/TG culture system was optimized by the critical inclusion of Vitrolife Ovoil™. We verified the excellent performance of our validated culture system by calculating our cumulative fresh embryo transfer rate between 2010–2013. The live birth rates by mean embryo numbers transferred for women using donor eggs who were <35 years old, 35–37 years old, 38–40 years old, and 41–42 years old were, respectively, 76.6%/1.89, 51.7%/2.21, 44.8%/2.23, 30.9%/2.49 and 19.9%/3.05. The age related implantation and live birth rate trends typically observed with the fresh ET (Figure 2A) were eliminated by applying FET often associated with PGT-A determinations (Figure 2B). Overall, the FET live birth rates by mean blastocyst number transferred for women using donor eggs, <35 years old, 35–37 years old, 38–40 years old,

and 41–42 years old were 54.5%/1.29, 67.7%/1.18, 63.8%/1.27, 60.2%/1.29, and 59.2%/1.26, respectively. The percentage of FET cycles using euploid blastocysts increased from 48.5% in 2013 to 86% in 2016, with an overall post-warming survival rate of 98% over 4 years. It is intriguing to note that the cumulative spontaneous abortion rates (SABs) did not differ between fresh ET and FET cycles involving patients under 38 years old (10.6% and 10%, respectively), while women over 38 years old experienced higher SABs with fresh ET (19.7%) as expected compared to FET patients (12.2%).

**Table 4.** Prospective randomized culture media comparison (G1.5/2.5 vs. mLG) assessing blastocyst quality and pregnancy outcomes between two age groups, with a secondary crossover experiment using the Ovoil™ on mLG medium microdrop dishes.

| Experiment A | | | | | | |
|---|---|---|---|---|---|---|
| Age/Treatment | #Pts | #/BLET | +β-hCG (%) | Clinical Preg. (%) | Live Birth (%) | #BL AA-BB (%) * |
| ≤34, G1/2.5 | 26 | 2.1 | 20 (77) | 20 (77) [a] | 19 (73) [a] | 142/285 (50) [a] |
| ≤34, mLG | 24 | 2 | 17 (71) | 13 (54) [b] | 11 (46) [b] | 149/241 (62) [b] |
| 35–40, G1/2.5 | 17 | 2.4 | 11 (65) | 9 (53) | 8 (47) | 58/149 (39) |
| 35–40, mLG | 23 | 2.4 | 16 (70) | 13 (57) | 11 (48) | 86/203 (42) |
| Experiment B | | | | | | |
| ≤34, mLg, + Ovoil™ | 33 | 2 | 27 (82) | 24 (73) | 22 (67) | 297/418 (71) [a] |
| 35–40 mLG, + Ovoil™ | 28 | 2.6 | 20 (71) | 19 (68) | 16 (57) | 95/162 (60) [b] |

G1/2.5-Vitrolife sequential G-media + Ovoil®; mLG-modified Global® + LG-LMO; LMO-light mineral oil; * BL development = BL-ET + BL-cryo (Blast ETs only); [a,b] column values within age groups are different ($p < 0.05$).

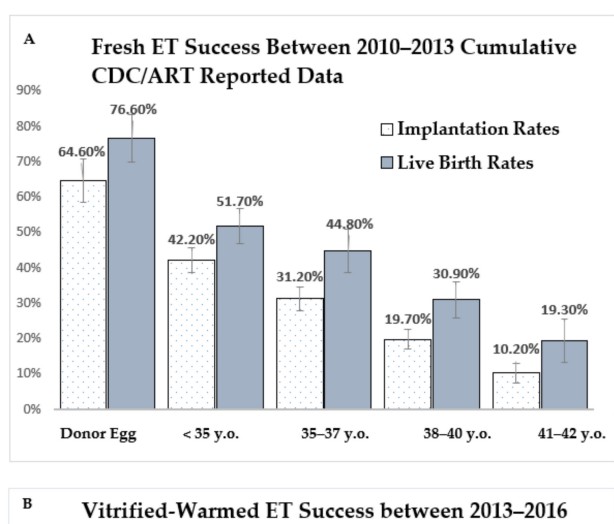

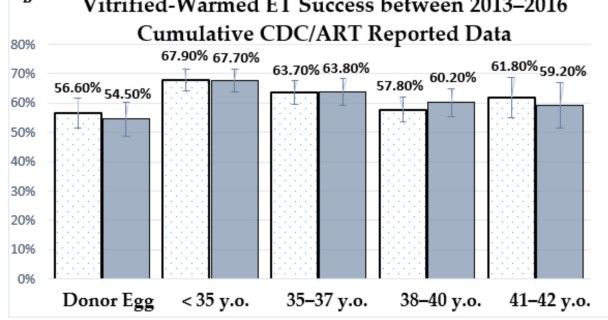

**Figure 2.** Accumulation of CDC/ART Surveillance Report Results from three physician clinics using our mLG/TG embryo culture system at the SCIRS/OF-NB ART Lab between 2010 thru 2016, verifying blastocyst viability following fresh (N = 1218 cycles; 2010–2013; Panel (**A**) and vitrified-warmed embryo transfer (N = 1808 patients; 2013–2016; Panel (**B**)) cycles.

The affiliate clinics were FCARE in Brea, CA; Fertility Center of Southern California (FCSC) in Irvine, CA; and Southern California Center for Reproductive Medicine (SCCRM) in Newport Beach, CA. About 12% of the data set population included donor egg cycles. Data is expressed as combined means ± 95% C.I. standard deviations.

## 4. Discussion

In the 21st century, the field of reproductive medicine witnessed improvements in culture media formulations and a growing menu of commercial options for laboratory application. A big emphasis was initially placed on sequential media formulations, as advocated by David Gardner [18], aiming to meet the specific physiological needs of the embryo at different developmental stages (early cleavage vs. post-compaction). Meanwhile, KSOM$^{AA}$ medium entered the Life Global marketplace as Global$^®$ medium (LG; [48]), which was optimized as a single use medium that worked under the principle that embryos will select what they need at changing developmental stages. These diverging concepts continue today, although the increase acceptance of single-step media, especially in conjunction with time lapse imaging [49,50], has spurred more companies to develop additional single use media. Both Gardner and Lane [19,21,25], as well as Biggers and associates [34,48,51] provided us with sound science supporting their formulations and aimed to minimize embryo stress through consistency in amino acid and antioxidant supplementation, as well as using stable dipeptides of L-glutamine that minimized ammonium waste-product production [26,34,35,52]. Knowing that the original sequential media provider (Vitrolife, G-series1/2 media) and single-step media supplier (LifeGlobal$^®$, LG medium) applied good science and superior quality control practices in the development, formulation, and manufacturing of their products, we focused our development of an optimized embryo culture system to these products.

While new media formulations strived to reduce embryo stress and improve in vitro development in culture [22], others addressed these issues from the perspective of incubator systems providing a physiologic gas environment (2–8% $O_2$; [53,54]) using TG conditions. Well studied in animal models [22,55], TG incubation offered advantages of reducing oxidative stress that could benefit the potential viability of in vitro produced human blastocysts with improved implantation and live birth success [56]. We observed similar significant trends in Expt.1 where embryos grown under physiologic TG conditions experienced greater implantation maintenance with more live births. Although the large capacity Hearus 240 incubators proved equally effective, the industry was moving toward smaller incubators for human embryo use that would have faster recovery intervals to better maintain equilibrium gas and pH conditions. While the first generation table-top incubators (MINC, Cook Medical) were being marketed, we chose to invest our future in new mini-Sanyo incubators. Similar to table-top incubators, the mini-box style aimed to reduce the number of patients per incubator (typically 1 or 2/incubator chamber). Combined with our elimination of Day 2 and Day 4 embryo checks, we have been better able to maintain optimum, equilibrium environmental conditions (gas and temperature). Being more traditional in design (box-style) and in their environmental control of gas conditions ($CO_2$ and $O_2$ set point variation, humidification), we proved in Expt. 2 that the Sanyo MCO-5M incubators were an excellent choice with proven clinical performance. These space efficient incubators did an excellent job maintaining their set temperature and %$O_2$ demands, recovered %$CO_2$ relatively quickly (within 15 min), and possessed a reliable electronic touch pad offering ease of use for calibration and % gas concentration adjustments. Furthermore, the inCu saFe™ stainless steel and SafeCell™ UV components of the design provided unprecedented contamination control features that were highly effective and not harmful to embryo development. Over a decade later, these incubators ($n = 18$) continue to function with consistency and reliably, proving to be relatively low maintenance with only occasional repairs needed.

As media products and incubator systems improved, so too did dependable in vitro blastocyst production. Most IVF programs experienced increases in implantation and live rates by converting

to Day 5 ET [20,52,56,57], being significantly better than Day 3 ET as confirmed in our early analysis (Table 3). We experienced consistently good blastocyst production and high pregnancy outcomes with mLG, being equal to or better than sequential media treatments, consistent with the reported results of several other groups [51]. Furthermore, we confirm that a single-step media culture system offers distinct advantages in quality control, ease of dish set-up, and reduced staff labor while minimizing the risk of error that could adversely impact pregnancy success [51]. Following the optimization of our blastocyst culture system with mLG/TG incorporating Ovoil™ (Expt. 4), we were even more confident to strive toward a practice of single embryo transfer as advocated by Gardner and Lane [52] years earlier. Certainly, the concurrent development, validation, and implementation of highly reliable vitrification [44,45] and blastocyst biopsy [41,43] procedures has allowed us to confidently implement single ET with unprecedented success, as reflected in the outcome verification of our FET cycles in Figure 2B. Note that implantation success and live birth rates are similar within age groups and each occurs at an equally high level across age groups when transferring a mean of 1 to 1.2 blastocysts/FET. This is atypical of the characteristic pattern observed with fresh ET (Figure 2A), where two or more embryos were routinely transferred between 2010–2013 resulting in lower implantation success relative to the higher live birth rates achieved per ET. The efficiency of embryo usage in FET cycles (Figure 2B) is so dramatic that the older age group uterine recipients (i.e., donor egg, 41–41 y.o. groups), who are susceptible to more spontaneous fetal losses after 8–10 weeks of gestation, actually experienced lower mean live birth rates than implantation rates. Another interesting observation involving the donor egg group in Figure 2B is that they historically exhibit reduced FET outcome trends compared to fresh ET, being more susceptible to repeat FET patient implantation failure bias. Those same donor egg recipients may fail a single fresh ET, but multiple FETs in a given year. In addition, the lower patient number in this category tends to exaggerate the inefficiency of success. Today, we have continued to adopt a "one embryo-one baby" mentality for patient management aimed at single, healthy term live births as reported by the CDC [47].

Under Gardner's advisory direction, Vitrolife provided other high-quality innovative and complementary products like paraffin oil (Ovoil™) and hyaluronan (HA, 500 μg/mL; EmbryoGlue™) aimed to optimize embryo culture and blastocyst implantation post-ET. Unique to Vitrolife and their product patents, they also supplemented their G3 and G5 media with HA (125 μg/mL) based on validation studies in the mouse model [37], which no other commercial company offered. The addition of HA was logical as it is a ubiquitous glycosaminoglycan found in the extracellular matrix, particularly noteworthy in follicular, oviductal, and uterine fluids. As a cell surface macromolecule, HA has a high affinity for binding water molecules that can form a lubricating viscous milieu and aid in the hydration of plasma membranes. Interacting with other proteins (e.g., hyaladherins), HA is believed to promote cellular differentiation and proliferation, gene expression regulation, and cell migration during implantation [37,58,59].

One of the disadvantages of commercial media requiring FDA-GMP approvals for distribution is that there is a reluctance by manufacturers to modify and improve formulations due to significant costs to the company. There is often a mentality of just sticking to what is already selling. Based on our interests to optimize our culture media, we identified and added a highly purified recombinant, high-molecular weight HA product (EAS) which we felt had significant potential merits as a serum-free supplement to LG (mLG). Our studies validate that this EAS was indeed not harmful and is more likely beneficial based on the outstanding implantation and live birth rates we achieved, which has attained elite levels (i.e., upper 5 percentile) as previously demonstrated by our lab [41,44,45]. In addition to its published role as an EAS which enhances implantation [60], we believe strongly that HA promotes membrane plasticity, aiding in cellular healing and protective functions associated with stressful events like trophectoderm biopsy and blastocyst vitrification. Certainly, other factors, like growth factors and cytokines, are undoubtedly involved in the dynamic cellular events mentioned above, and that is why we also preferred the use of synthetic protein supplements (SPS) instead of a defined HSA [61,62].

The α & β globulin fraction (14–16% by volume) is believed to contain growth promoting factors that supplement the beneficial qualities of albumin alone [31,61].

Overall, we feel we have effectively enriched a superior single-step media (KSOM$^{AA}$, i.e., Global$^®$; mLG) methodically developed by Dr. Biggers and his colleagues [6,51] for optimum human in vitro embryo culture usage. Meanwhile, we have strived to employ the culture philosophies of Gardner and Lane, aiming to minimize embryo stressors in their microenvironment by providing quality, consistency, and stability in media and macromolecule products [18,63,64]. By reducing the albumin content of mLG medium (i.e., adding only 7.5% SPS), we allowed ourselves to add a concentrated EAS stock solution (5 mg/mL) without diluting the clinical formulation of LG. Combined with a low oxidative, humidified incubation environment for embryos grouped cultured in microdroplets under a high quality, chemically stable paraffin oil we have sought to minimize cellular stress to membranes and organelles and promote more viable blastocysts and fewer miscarriages. Although we have successfully developed and validated our embryo culture system, optimization of blastocyst production, implantation and healthy live singleton births is only achieved when strict adherence to total quality control management is maintained in all facets of the IVF process.

It is important to mention that our success rates and experiences may not translate to the results generated by other clinical laboratories, as culture media comparisons are subject to over 200 variables [65]. We acknowledge there are inherent variations and correlations between physician effects and laboratory outcomes. Yet, the Phase 2 verification data did integrate the experience of three different clinics reporting to the CDC, as to not present biased success of a single top performing physician. Though our individual studies may lack statistical power, they are in alignment with the global experimental recommendations of Pool, Schoolfield and Han [65] in that our validation of a mLG/TG culture system is based on a sufficient caseload of diverse infertile patients over several years with repeatedly high success rates in a laboratory adhering to a strong quality assurance processes. Indeed, progress and continued success with ART applications requires a dedicated team of individuals who think outside the box, apply good sound scientific principles, and adhere to a strict quality control program.

## 5. Conclusions

By systematically evaluating and implementing various components of an embryo culture system, including TG conditions, paraffin oil and the addition of endotoxin-free EAS and SPS, we were able to optimize our embryo development while supporting cellular membrane wellness under stressful in vitro conditions (e.g., culture, cell biopsy, vitrification). Overall, our mLG/Ovoil™/TG culture system has proven to be effective creating reliably high blastocyst production, implantation and healthy live births.

**Author Contributions:** The lead author (M.C.S.) was responsible for the conceptualization of the project and writing/editing of the manuscript. In addition, he and S.Z. were responsible for performing the prospective validation experiments and the written and oral presentation of the data at scientific conferences. Together with two other senior Embryologists (N.L.N., J.B.W.), they worked together as a team to evaluate embryos, perform ART procedures and collect & process data. Meanwhile, the three physician authors (R.E.A., I.H., C.T.L.) contributed patients to these studies, being fully supportive of our laboratory decisions and the publication of this manuscript. All authors have read and agreed to the published version of the manuscript, having contributed to the results reported.

**Funding:** No external source of funding.

**Acknowledgments:** We are grateful to the dedicated, hardworking Embryology staff at the Southern California Institute for Reproductive Sciences (SCIRS), before it became Ovation Fertility, whom helped contribute to the exceptional success we are continuing to achieve.

**Conflicts of Interest:** The authors declare no conflict of interest. The funders had no role in the design of the study; in the collection, analyses, or interpretation of data; in the writing of the manuscript, or in the decision to publish the results.

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
