# Peer review of "Systematic Development, Validation and Optimization of a Human Embryo Culture System"

_2673-3897, doi:10.3390/reprodmed1010001_

Round 1

Reviewer 1 Report

I consider this study deals with an interesting topic in reproductive biology and it could be helpful in improving human embryos in vitro culture systems. In general terms, experimental design is appropriate and materials and methods used are adequate. Introduction and discussion sections have an appropriate size and adequate bibliography revision. Therefore, I consider this manuscript could be published after some few minor changes that could improve its quality:

Please, include “n” in all the experiments described in materials and methods section and in table and figure legends It is important that in both, tables and figures, standard deviation or error were showed Please, modify figure 2 in order to represent standard deviation or error as well as statistics It would be helpful to improve the flow chart Considering the different approaches and experiments carried out in this manuscript I recommend to focused the summarized main conclusions of the study at the end of the discussion

Author Response

Reprodmed – Manuscript #697336

Author response to Reviewer Comments

Even before we received the Reviewers comments I had already initiated corrective actions to typos and an incomplete sentence in the text. With your assistance we have revised all sections of the text.  Note, we have had difficulty inserting the revised Fig.2, so it has been placed at the end of the paper for the Editorial office to professionally integrate it.  As advised by the editor, we are providing the reviewers with our text corrected red-lined version to see exactly what we did.  We readily admit that this paper is not intended to be a statistically significant comparison of media, and we have made a point not to draw any comparisons between Labs.  Instead, our purpose was to exhibit our justified validation of a culture system that we have optimized for our use. In light of a growing FDA presence, it is important to validate and publish these processes.   We hope the reviewers are satisfied with our revised version.

Rev# 1

We have made sure the “number of patient cycles” is revealed in all the Experiments of the M&M Experimental Design section.  In addition, to further clarify an inference by Review#2 regarding donor egg cycles, we have added the “n” delineating egg donor cycles.  This included a superscripted addition in Table 1.  As far as the Fig. 2, we have added the combined mean over each column +/- the 95% CI standard deviation.

Reviewer 2 Report

I have read with interest the report by Schiewe and colleagues entitled "Systematic Development, Validation and Optimization of a Human  Embryo Culture System".  In essence this work simply describes implementations of new media and incubators that provide a low oxygen culture environment plus the the influence of adding hyaluronan to an existing culture formulation.  The authors use outcomes from donor egg cycles submitted to the CDC as validation data and show, in a stepwise fashion, incremental improvements tp those outcomes.  I have the following comments:

1.  The pitfalls of extrapolating experiences from a specific culture system, mainly media, from one laboratory to another has been examined in detail by Pool, Schoolfield and Han (2012).  If the authors are implying this is possible, data from a separate center where outcomes improved after instituting this optimized system, need to be presented.  The number of variables resident in a given IVF cycle, many arising in the clinic but impinging upon laboratory performance, often preclude this.  The paper does have value for publication in that it details an exercise that most centers have not performed but must, given the pivotal role the embryo culture systems will play in providing optimized outcomes with the advent of machine learning and big data that will feed those algorithms in the very near future.

2.  All data are derived from donor cycles.  How does this system perform in the general infertility population?

3.  Quinn, Kerin and Warnes produced Human Tubal Fluid medium but that medium was not devoid of phosphate. The first phosphate-free medium used for human embryo culture was P-1 and the unpublished results from its use spurred Quinn to develop a phosphate-free modification of HTF in the 1990's.

Minor points - 

1.  Either include all academic degrees or no academic degrees in the author list.

2.  Line 295 refers to our field as the "IVF industry".  I know of no other medical or surgical discipline or subdiscipline that is referred to as an 
industry", not even cosmetic surgery.  Replace it with a professional descriptor.

Author Response

Reprodmed – Manuscript #697336

Author response to Reviewer Comments

Rev# 2

We greatly appreciate the opinion and assistance of this reviewer.  The reference provided was unfamiliar to us but was definitely appropriate to add to end of our discussion.  Granted it could probably be integrated earlier in the discussion or even the introduction but in the end I was not certain its impact or importance is any less in conclusion.   We appreciate being able to warn the reader as to the risk of comparative inference as it applies to comparative media trials.  Indeed, I believe we have stressed that the purpose of this “validation” study is to verify the effectiveness of a “culture system” (medium, additives and incubator environment) that is optimized for our laboratory.  The Pool, Schoolfield and Han (2012) ref is perfect to educate the reader to the pitfalls of comparative media trials, and we thank them for this gem! We are confused by this statement, as only a normal fraction of the data presented is derived from donor egg cycles. I have made a point of clarifying the % of cycles involving donors.  In fact, we also clarified that the prospective evaluation in Expt. 4 actually excluded donor eggs stating  that “90 autologous patient cycles” were…..    .  We hope this helps clarify the issue Thank you for this clarification in regards to my misinterpretation of the HTF formulation. I knew P-1 was a significant improvement which I did adopt as my fertilization medium for over a decade. Academic degrees were added, but will leave it up to the editor as to what is an acceptable format for this new online journal. I would argue that the” field of Reproductive Medicine” is becoming more of an industry every year with its ever growing commercialism and corporate influences. I have toned down and redirected the inference to “industry”